# Seroprevalence of Hantavirus in Forestry Workers, Northern France, 2019–2020

**DOI:** 10.3390/v15020338

**Published:** 2023-01-25

**Authors:** Catarina Krug, Emma Rigaud, Dieyenaba Siby-Diakite, Laetitia Bénézet, Pavlos Papadopoulos, Henriette de Valk, Gaëtan Deffontaines, Alexandra Septfons, Jean-Marc Reynes

**Affiliations:** 1Santé Publique France, 94410 Saint-Maurice, France; 2European Centre for Disease Prevention and Control (ECDC), 169 73 Solna, Sweden; 3Caisse Centrale de la Mutualité Sociale Agricole, 93000 Bobigny, France; 4Institut Pasteur, Université Paris Cité, 75015 Paris, France

**Keywords:** France, seroepidemiologic studies, occupational exposures, Hantavirus, Puumala virus

## Abstract

We aimed to estimate the seroprevalence of *Puumala orthohantavirus* (PUUV) among forestry workers in northern France, and to explore sociodemographic risk factors. We conducted a random cross-sectional seroprevalence survey among 1777 forestry workers in 2019–2020. The presence of immunoglobulin G against PUUV antigens in serum was assessed using enzyme-linked immunosorbent assay and confirmed using immunofluorescence assay. Poisson regression models were used to explore factors associated with seropositivity. Weighted seroprevalence was 5% (3–6) in northeastern France, 4% (2–6) in north central France, and 1% in two regions located in the center of the country (Auvergne and Limousin). There were no seropositive workers detected in northwestern France. Seropositivity was associated with age, sex, and cumulative seniority in the forestry sector. Seroprevalence was highest in known endemic areas of the northeast and lowest in the northwest. Nevertheless, we found serological evidence of PUUV infection in two regions located in the center of the country, suggesting circulation of the virus in these regions, previously thought to be non-endemic.

## 1. Introduction

Zoonotic hantaviruses are rodent-borne enveloped viruses with a tri-segmented, negative sense, single-stranded RNA genome, belonging to the *Orthohantavirus* genus (Subfamily *Mammantavirinae*, Family *Hantaviridae*, Order *Bunyavirales*). They are responsible in Eurasia for mild to severe hemorrhagic fever with renal syndrome (HFRS) [1].

*Puumala orthohantavirus* (PUUV) is the hantavirus species most commonly detected in patients in metropolitan France, whereas *Seoul orthohantavirus* (SEOV) and *Tula orthohantavirus* (TULV) are detected sporadically. As in some other European countries [1,2], the spatial distribution of HFRS cases is not homogenous in metropolitan France, with detection of human cases being restricted to the northeastern area, including the highly endemic regions of Nord-Pas-de-Calais, Picardie, Franche-Comté [3,4]. There were less than ten individuals with anti-hantavirus antibodies (probable SEOV cases) reported outside the northeastern area in the nineties [5]. The main risk factors for human PUUV infection include living near the forest, the presence of rodents in the dwellings, and leisure or professional activities in the forest [6].

In contrast to the spatial distribution of HFRS cases, the main natural hosts of these three hantavirus species are present in most regions of metropolitan France. Hantavirus hosts include the bank vole (*Myodes glareolus*) for PUUV, the brown rat (*Rattus norvegicus*) for SEOV, and the common vole (*Microtus arvalis*) for TULV. There are various hypotheses to explain this heterogeneous geographic distribution of PUUV, including environmental factors (influencing the rodent population dynamics and virus survival outside the host), rodent susceptibility (including virus replication and excretion), genetic characteristics of the viruses (favoring or not, its replication, its excretion, its survival outside the host), and human behaviors (contact with rodents or their excreta) [2,3,7]. PUUV prevalence studies in bank vole populations have only been conducted in the northeastern quarter of metropolitan France, in the endemic area, and its vicinity [8,9,10,11,12,13,14,15,16,17]. Bank voles have been found infected in these areas, with varying prevalence level of anti-PUUV antibodies according to time and space. There is no information outside this area.

Cases may be missed outside the northeastern quarter of metropolitan France because patients are not frequently tested for hantavirus [18]. The occurrence of the disease is not expected by clinicians and awareness of hantavirus diseases is likely to be low among clinicians in non-endemic areas [18].

There is little information on the presence of PUUV in French regions that are considered non-endemic. Given that forestry workers are at higher risk than the rest of the population due to potential exposure to rodents, particularly the bank vole, host of PUUV, and their excreta [19], conducting a study targeting forestry workers can help us document the distribution of hantavirus. Our primary aim was to estimate the overall and regional prevalence of antibodies against PUUV antigens among forestry workers in northern France. Our secondary aim was to compare seroprevalence between geographic areas, particularly northeast (considered to be endemic) and other northern areas (considered to be non-endemic); and to determine sociodemographic factors associated with hantavirus seropositivity.

## 2. Materials and Methods

### 2.1. Sample Size Calculations

The original study was designed to investigate the seroprevalence of *Borrelia burgdorferi* sl. The sample size for our study was therefore determined to answer this research objective. The number of required subjects was calculated for each of the following five geographic areas: northwestern France including Normandie (Haute-Normandie and Basse-Normandie), Bretagne and Pays-de-la-Loire regions; north central France including Hauts-de-France (Picardie, Nord-Pas-de-Calais), Île-de-France, and Centre regions; northeastern France including Alsace, Lorraine, Champagne-Ardenne, Bourgogne, and Franche-Comté regions; and Auvergne and Limousin regions, both located in the center of France. Assuming a *B. burgdorferi* sl seroprevalence of 5% in northwestern and north central France and of 15% in northeastern France [20,21,22], a sample of 2591 individuals would be required to achieve a minimum precision of 1.5% and a maximum precision of 4% in the estimation of *B. burgdorferi* sl seroprevalence. A total of 1778 forestry workers were included from May 2019 to March 2020. The number of participants was lower than required due to the COVID-19 pandemic as the lockdown led to the interruption of data collection. We conducted power calculations to estimate the precision of hantavirus seroprevalence estimation using the available sample (see Appendix A).

### 2.2. Study Design, Participants, and Samples

We conducted a random cross-sectional seroprevalence survey in northern France from May 2019 to March 2020. The target population consisted of all forestry workers aged 18 or older, working at least once a week in a wooded environment or park covered by the French farmers’ insurance fund (Mutualité Sociale Agricole, MSA) of one of the following administrative regions: Normandie (Haute-Normandie and Basse-Normandie), Bretagne, Pays-de-la-Loire, Hauts-de-France (Picardie, Nord-Pas-de-Calais), Île-de-France, Centre, Alsace, Lorraine, Champagne-Ardenne, Bourgogne, Franche-Comté, Auvergne, and Limousin.

To recruit study participants, we used a stratified random sample. Strata were defined by region (as defined by affiliation to the local MSA insurance fund), occupational activity (silviculturist, forestry technician, woodcutter, etc.), and working status (self-employed, salaried, other). Within each stratum, simple random sampling was used to select individuals. The number of individuals sampled was proportional to the size of the strata.

Individuals were invited for a medical visit by their occupational health physician. After obtaining informed consent, participants were interviewed, and blood samples were collected. The questionnaire administered included sociodemographic information such as age and region of MSA insurance fund affiliation, as well as details on occupational activity including main profession, average weekly exposure to the forest, and cumulative seniority in forestry activities. The questionnaire also included questions on the main departments where forestry activities occurred during the last 12 months, as well as secondary departments or countries. All blood samples were anonymized at the time of collection by giving each a unique number. Blood samples were kept at room temperature (10–30 °C) and centrifuged (3200 rpm for 10 min, at 20–24 °C) within 48 h. Sera were frozen at −20 °C within 2 h after centrifugation. Serum was divided into five aliquots and stored at −30 °C until analyzed.

### 2.3. Serological Methods

Collected sera were screened for the presence of anti-PUUV immunoglobulin G (IgG) at 1:100 dilution using homemade enzyme-linked immunosorbent assay (ELISA) according to Rossi and colleagues’ method [23]. ELISA-positive sera were screened at 1:64 dilution using a homemade immunofluorescence assay (IFA) according to Niklasson and colleagues’ method [24], to confirm the presence of anti-PUUV IgG. Both assays were conducted under ISO 15,189 accreditation. The Hantavirus National Reference Center has estimated a sensitivity of 100% and a specificity of 94.8% for its ELISA and of 100% and 98.1% for its IFA for serum samples from patients taken 10 days after symptom onset (Jean-Marc Reynes, personal communication). A negative result indicated a lack of PUUV exposure while a positive result indicated hantavirus exposure but not necessarily specific to PUUV, due to serological cross-reaction with other hantaviruses [25].

### 2.4. Statistical Analysis

All statistical analyses were weighted. The weights used considered the sampling weight and were adjusted for non-response. Non-response was corrected by reweighting using the equal-quantile score method [26], based on socio-demographic and geographic data and the professional activity sector available in the sampling frame for all individuals. A calibration by raking ratio method was then applied, using the distributions by sex, age group, and professional activity sector in the target population, using the SAS macro Calmar [27]. 

We described population demographic characteristics with absolute and relative frequencies for categorical variables and with mean and standard deviation for continuous variables. We explored factors associated with hantavirus seropositivity by estimating seroprevalence ratios and 95% confidence intervals (CI) using weighted Poisson regression models with robust standard errors. The clustering of observations by geographic area was accounted for using a fixed area effect in all models. The outcome of interest was hantavirus seropositivity, and we used one model per factor of interest (the main exposure variable). Causal diagrams were made between hantavirus seropositivity and each factor of interest using DAGitty ([28,29]; diagrams not shown). Based on those causal diagrams, it was deemed reasonable to use unconditional models for the geographic area, age, and main profession (i.e., no important confounders were identified for those factors). Also, we identified age as a confounder of the association between seniority and hantavirus seropositivity; and main profession as a confounder of the association between average weekly exposure time in forest and hantavirus seropositivity. Nevertheless, the strong collinearity between age and seniority (Pearson correlation coefficient of 0.76), precluded a multivariable analysis testing both factors. Analyses were performed using Stata 14.

## 3. Results

Out of 4484 forestry workers randomly selected, 1778 participated in this study (response rate of 41% [30], and serological test results were available for 1777. Demographic characteristics of the study population are described in Table 1. The mean (range) age of participants was 44 (18–87) years, 94% (1714/1777) were men, and one-third of participants were affiliated to three French farmers’ insurance funds regions in northeastern France: Lorraine (13%, 202/1777), Champagne-Ardenne (12%, 148/1777) and Alsace (10%, 133/1777). As for the characteristics of the work performed, the majority of participants were working in one of four occupational activities: woodcutter (34%, 622/1768), forest technician or ranger (21%, 357/1768), forestry machine operator (18%, 319/1768), and silviculturist (17%, 289/1768). Seventy-eight percent (1399/1758) of participants were exposed to the forest, forest borders or edges, parks, and wooded gardens for longer than 20 h a week (Table 1).

Out of 1777 individuals, 50 (2.8%) had IgG against PUUV antigens in their serum. We estimated a weighted seroprevalence of 3.3% (95% CI: 2.5–4.3). Seroprevalence varied across age groups, being lowest among individuals aged 18–40 years (1%, 95% CI: 0.73–3) and highest among those older than 60 years (6%, 95% CI: 3–11). Specifically, seroprevalence was four times higher (95% CI: 1.5–10.2) among individuals older than 60 years than among those aged 18–40 years (Table 2). All workers with anti-PUUV IgG were male. In Figure 1, we observe a seroprevalence increasing from northwest to northeast France, with no seropositive workers detected in northwestern France, a low seroprevalence in Centre (1%, 95% CI: 0.25–3), Limousin (1%, 95% CI: 0.09–3) and Auvergne (1%, 95% CI: 0.33–4) regions, and highest seroprevalence in Hauts-de-France (7%, 95% CI: 4–12) and Franche-Comté regions (8%, 95% CI: 5–14; Table 2). Workers with anti-PUUV IgG in Auvergne and Centre reported having been only exposed in their own region, with the exception of one worker in Auvergne who also traveled to Limousin and Centre. The worker with anti-PUUV IgG in Limousin was not exposed in other French regions, but traveled to Eastern Europe/Balkans, hantavirus endemic areas. There was no association between traveling to neighboring countries, and having anti-PUUV IgG.

Seroprevalence appeared to be greater among forest technicians and forest rangers (6%, 95% CI: 4–9) than among woodcutters (3%, 95% CI: 2–5) or silviculturists (2%, 95% CI: 0.63–4), but differences were not statistically significant (Table 2). Seroprevalence increased with average weekly exposure time to the forest, forest borders or edges, parks, and wooded gardens, with a seroprevalence of 1% (95% CI: 0.32–4) among workers exposed for less than 10 hours a week and 4% (95% CI: 3–5) among workers exposed for longer than 20 h a week. Corresponding to a prevalence ratio of four (95% CI: 1.2–12.5) between the two categories after adjusting for the main profession. Seroprevalence was lower among forestry workers with one to ten years of work in the sector (1%, 95% CI: 0.50–2) than among those with over 20 years of work in the sector (5%, 95% CI: 3–8). Specifically, seroprevalence was five times higher (95% CI: 1.9–12.5) among individuals with 21–30 years of work than among those with 1–10 years of work in the sector (Table 2). There was no association between hunting, fishing, hiking, or harvesting, and having anti-PUUV IgG.

## 4. Discussion

Our primary aim was to estimate the overall and regional seroprevalence of the hantavirus among forestry workers in northern France. In our study, the overall seroprevalence for hantavirus was estimated at 3.3% (95% CI: 2.5–4.3), which aligns with a previous meta-analysis by Riccò and colleagues [19] that indicated a seroprevalence of 4.1% (95% CI: 2.7–6.1) among European forestry workers. Nevertheless, results may not be directly comparable to ours, because this metanalysis included data from various European countries (west, south, and east European countries) with heterogeneous hantavirus incidences, and a variety of laboratory methodologies.

Our secondary aim was to compare seroprevalence between geographic areas, particularly northeast and other northern areas. As anticipated, the seroprevalence among forestry workers was greatest in known endemic areas of northeastern and north central France, particularly, in the highly endemic regions of Franche-Comté and Hauts-de-France [18]. Champagne-Ardenne region is also a known endemic region. We observed lower seroprevalence in the Champagne-Ardenne region (4%) than a previous study conducted in the Ardennes (part of the Champagne-Ardenne region) by Penalba and colleagues [31], who described a 15% seroprevalence in 55 forestry workers [31]. No details on the laboratory methods used in their study were described, which makes it difficult to compare with our results. Île-de-France, an endemic area with a lower number of cases than other endemic areas, had a lower seroprevalence of 3%, in accordance with a previous study conducted in 1991–1992 among forestry workers [32].

We did not find evidence of the circulation of PUUV in northwestern France (Normandie, Bretagne, and Pays-de-la-Loire), which is considered non-endemic [18]. Specifically, given that there are around 1000 forestry workers in northwestern France, and that we found zero hantavirus workers with anti-PUUV IgG among the 396 tested, we estimate at a 95% confidence level that the real seroprevalence of hantavirus in that area is below 1.3% [33]. This is in agreement with the absence of detection of hantavirus cases by healthcare practitioners [18]. In contrast, we found evidence of hantavirus infection (three workers with anti-PUUV IgG) in the Auvergne and Limousin regions, located in the center of the country and previously considered to be non-endemic [18]. A landscape and regional environmental analysis of the spatial distribution of hantavirus human cases in Europe showed that HFRS cases should be expected in these areas [2]. Similarly, a previous study performed in the 80′s in the Auvergne region (located in the center of France) had 2.3% of more than 1000 farmers testing positive for IgG against Hantaan orthohantavirus or PUUV antigens by immunofluorescence (serum dilution starting at 1:16) [34]. One of the limitations of our study was that we were unable to estimate hantavirus seroprevalence across the whole of France. Now that our study found evidence of hantavirus infection in workers in an area previously considered to be non-endemic, we recommend closer monitoring of southern France regions, as the area of virus circulation could very well extend. Nevertheless, we cannot dismiss the hypothesis that the individuals with anti-PUUV IgG in Auvergne and Limousin did not recall being exposed to endemic regions or countries (recall bias).

Finally, we also aimed to determine sociodemographic factors associated with the prevalence of anti-PUUV IgG. Age was positively associated with the presence of anti-PUUV IgG similar to the results reported in other studies [35,36]. This is explained by the fact that exposure to hantavirus accumulates over the course of a lifetime, due to repeated exposure to rodents infected excreta, and by the long half-life of these antibodies [37]. Similar to the results reported in other studies [35,36], seroprevalence also increased with longer exposure time per week in the forest, forest borders or edges, parks, and wooded gardens and with cumulative seniority in the forestry sector. The proportion of women in our target population was low, which affected our ability to draw conclusions on the relationship between sex at birth and the presence of anti-PUUV IgG.

Forestry workers are at higher risk of interacting with rodent hosts than the general population. Thus, our results cannot be generalized to the overall French population. We can also hypothesize that forestry workers are more aware than the general French population of occupational-related infections and their preventive measures. Indeed, there are specific HFRS prevention campaigns targeting forestry workers, as it is part of the list of occupational diseases in France [38]. Moreover, a previous French study indicated that forestry workers had greater knowledge about tick-borne diseases and were more likely to protect themselves than the general French population [39], indicating that they do follow recommendations and have a better knowledge of occupational diseases than the general population. Since the original study was aimed at investigating the seroprevalence of *B. burgdorferi* sl, we did not collect information on the use of preventive measures for hantavirus infections (such as the use of masks while working), and we could not evaluate their relationship with hantavirus seroprevalence.

In our study, we limited selection bias by randomly selecting forestry workers and by adjusting for non-response according to differences in sociodemographic and professional characteristics. Nevertheless, we cannot exclude the possibility of residual selection bias due to differences in hantavirus exposure between participants and non-participants with the same socio-demographic and professional characteristics.

## 5. Conclusions

In conclusion, the regional seroprevalence of hantavirus antibodies among forestry workers was greatest in the known endemic areas of northeastern and north central France, and lowest in the non-endemic northwestern France. Nevertheless, we found indications of the circulation of hantavirus in the Auvergne and Limousin regions, located in the center of the country, even though they were previously thought to be non-endemic. These results may indicate some risk of local HFRS cases in these regions, and the need to raise awareness among general and hospital practitioners and among at-risk populations. Practitioners in non-endemic areas should still be aware of the possibility of HFRS cases, especially among patients returning from endemic areas. An improved understanding of hantavirus infection rates in reservoir host species and virus transmission to humans is still needed to fully understand the absence of infection in non-endemic areas. Rodents’ ecology is highly variable, not only at the geographical level, but also over time, due to complicated interactions with their environment [40].

Routine surveillance has the disadvantage that cases might be missed in newly endemic areas that are still thought to be non-endemic [18]. The information obtained from routine surveillance may thus be complemented by sporadic zero surveillance, a useful tool to assess the cumulative exposure to hantavirus and the geographic distribution of human exposure. A reinforcement of sensitization campaigns among occupational health workers on hantavirus infection could help early identification of cases in endemic and non-endemic areas.

## Figures and Tables

**Figure 1 viruses-15-00338-f001:**
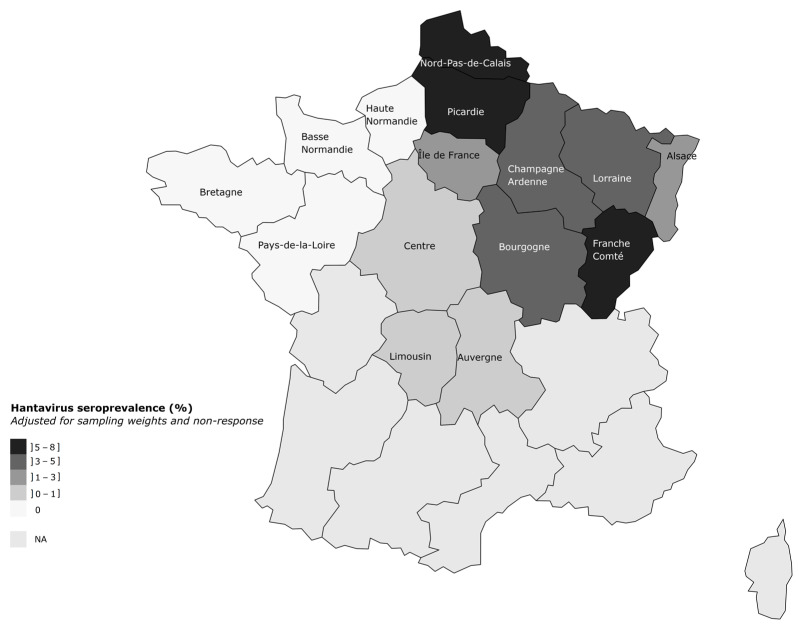
Adjusted prevalence of immunoglobulin G against *Puumala orthohantavirus* antigens among forestry workers in northern France, 2019–2020.

**Table 1 viruses-15-00338-t001:** Demographic features of forestry workers targeted in the *Puumala orthohantavirus* seroprevalence survey, northern France, 2019–2020.

Demographic Features		Weighted Distribution % (n/N)
Age in years, mean (min–max)		44 (18–87)
Age group, years	18–40	41% (702/1777)
	41–50	23% (430/1777)
	51–60	28% (520/1777)
	>60	8% (125/1777)
Sex	Male	94% (1714/1777)
	Female	6% (63/1777)
Administrative region ^1^	Alsace	10% (133/1777)
	Lorraine	13% (202/1777)
	Champagne-Ardenne	12% (148/1777)
	Bourgogne	6% (44/1777)
	Franche-Comté	9% (145/1777)
	Auvergne	8% (164/1777)
	Hauts-de-France (Picardie, Nord-Pas-de-Calais)	9% (143/1777)
	Île-de-France	3% (33/1777)
	Centre	9% (197/1777)
	Limousin	5% (144/1777)
	Normandie (Basse Normandie, Haute-Normandie)	3% (81/1777)
	Bretagne	4% (135/1777)
	Pays-de-la-Loire	7% (208/1777)
Geographic area ^2^	Northeast	51% (672/1777)
	North central	24% (401/1777)
	Northwest	12% (396/1777)
	Auvergne	8% (164/1777)
	Limousin	5% (144/1777)
Occupational activity	Silviculturist	17% (289/1768)
	Woodcutter	34% (622/1768)
	Forestry machine operator	18% (319/1768)
	Forest technician/forest ranger	21% (357/1768)
	Hunting technology/gamekeeper/fishery guardian	8% (144/1768)
	Gardener/landscaper	1% (20/1768)
	Other	1% (17/1768)
Average weekly exposure time in the forest, forest borders or edges, parks, and wooded gardens exclusively, hours	<10	10% (160/1758)
	10–20	12% (199/1758)
	>20	78% (1399/1758)
Cumulative seniority in forestry activities, years	1–10	27% (483/1777)
	11–20	25% (448/1777)
	21–30	22% (406/1777)
	31–40	17% (317/1777)
	>40	8% (123/1777)

^1^ Region of French farmers’ insurance fund affiliation. ^2^ Northeast includes Alsace, Lorraine, Champagne-Ardenne, Bourgogne, and Franche-Comté regions; North central includes Hauts-de-France, Île-de-France, and Centre regions; Northwest includes Normandie, Bretagne, and Pays-de-la-Loire regions. Note that Auvergne and Limousin regions are located in the center of France.

**Table 2 viruses-15-00338-t002:** Weighted prevalence and prevalence ratio of immunoglobulin G (IgG)against *Puumala orthohantavirus* (PUUV) antigens among forestry workers, northern France, 2019–2020.

Demographic Features		Number of Workers with Anti-PUUV IgG/Number of Workers Tested	Weighted Seroprevalence, (95% CI) ^3^	Weighted Prevalence Ratio ^4^	*p*-Value for the Prevalence Ratio ^5^
Age group, years	18–40	8/702	1% (0.73–3)	Reference	-
	41–50	12/430	4% (2–7)	3.2 (1.4–7.6)	0.008
	51–60	24/520	5% (3–7)	3.4 (1.6–7.1)	0.001
	>60	6/125	6% (3–11)	3.9 (1.5–10.2)	0.005
Sex	Male	50/1714	3% (3–5)	NA	
	Female	0/63	-	NA	
Administrative region ^1^	Alsace	2/133	2% (0.44–5)	NA	
	Lorraine	8/202	4% (2–7)	NA	
	Champagne-Ardenne	7/148	4% (2–9)	NA	
	Bourgogne	2/44	5% (1–17)	NA	
	Franche-Comté	13/145	8% (5–14)	NA	
	Auvergne	2/164	1% (0.33–4)	NA	
	Hauts-de-France (Picardie, Nord-Pas-de-Calais)	12/143	7% (4–12)	NA	
	Île-de-France	1/33	3% (0.50–16)	NA	
	Centre	2/197	1% (0.25–3)	NA	
	Limousin	1/144	1% (0.09–3)	NA	
	Normandie (Basse Normandie, Haute-Normandie)	0/81	-	NA	
	Bretagne	0/135	-	NA	
	Pays-de-la-Loire	0/208	-	NA	
Geographic area ^2^	Northeast	32/672	5% (3–6)	Reference	-
	North central	15/401	4% (2–6)	0.77 (0.43–1.4)	0.364
	Northwest	0/396	-	-	-
	Auvergne	2/164	1% (0.33–4)	0.26 (0.07–0.96)	0.044
	Limousin	1/144	1% (0.09–3)	0.11 (0.02–0.69)	0.018
Occupational activity	Silviculturist	4/289	2% (0.63–4)	0.49 (0.17–1.4)	0.190
	Woodcutter	15/622	3% (2–5)	Reference	-
	Forestry machine operator	8/319	3% (2–7)	1.1 (0.50–2.6)	0.763
	Forest technician/forest ranger	22/357	6% (4–9)	1.6 (0.87–3.1)	0.126
	Hunting technology/gamekeeper/fishery guardian	0/144	-	-	-
	Gardener/landscaper	0/20	-	-	-
	Other	1/17	5% (0.74–23)	1.5 (0.23–9.4)	0.674
Average weekly exposure time in forest, forest borders or edges, parks, and wooded gardens exclusively, hours	<10	2/160	1% (0.32–4)	Reference	-
	10–20	5/199	3% (1–6)	2.8 (0.71–10.8)	0.141
	>20	42/1399	4% (3–5)	3.8 (1.2–12.5)	0.026
Cumulative seniority in forestry activities, years	1–10	5/483	1% (0.50–2)	Reference	-
	11–20	6/448	2% (0.80–4)	1.6 (0.53–4.7)	0.408
	21–30	16/406	5% (3–8)	4.9 (1.9–12.5)	0.001
	31–40	14/317	5% (3–8)	4.2 (1.7–10.8)	0.002
	>40	9/123	7% (4–13)	5.4 (2.0–14.7)	0.001

PUUV, *Puumala orthohantavirus*, IgG, immunoglobulin G. ^1^ Region of French farmers’ insurance fund affiliation. ^2^ Northeast includes Alsace, Lorraine, Champagne-Ardenne, Bourgogne, and Franche-Comté regions; North central includes Hauts-de-France, Île-de-France, and Centre regions; Northwest includes Normandie, Bretagne, and Pays-de-la-Loire regions. Note that Auvergne and Limousin regions are located in the center of France. ^3^ Weighted seroprevalence. Seroprevalence is not adjusted for the sensitivity and specificity of the diagnostic test. ^4^ Calculated using weighted univariable Poisson regression models with hantavirus seropositivity (yes/no) as the outcome and the variable on the left as the exposure. The association between hantavirus and average weekly exposure time in forests, forest borders or edges, parks, and wooded gardens was adjusted for occupational activity. ^5^ The overall *p*-values were 0.007 for age, 0.025 for the geographic area, 0.056 for occupational activity, 0.218 for average weekly exposure time in the forest, and 0.001 for cumulative seniority in the forestry sector.

## Data Availability

Data is available upon request.

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
