# Peer review of "Seroprevalence of Hantavirus in Forestry Workers, Northern France, 2019–2020"

_viruses, 2023, doi:10.3390/v15020338_

Round 1

Reviewer 1 Report

I reviewed this interesting paper about seroprevalence of PUUV infection in french forestry workers. Although the original sample population was selected to assess exposure to Borrelia burgdorferi they made an appropriate subject selection for this second study. The laboratory methods are well described and are correct. I suggest giving more information in the introduction about prevalence studies in rodent reservoirs in the different regions of France; knowing the magnitude of the exposure help to understand the results better. It would also be informative to know if there are sequelae of HFRS concentrated in any particular group related to forestry workers. This information would give fundament to the main goal of the study.

 If there are no surveillance studies, propose them in the discussion. In addition, describe the dynamic of rodents in the areas where the researchers are interested and report if any human outbreak has occurred and where the associated risk factors are.

Finally, the information obtained contributes to understanding this zoonosis in risk populations and promotes recognition and prevention.

Author Response

R1 : I reviewed this interesting paper about seroprevalence of PUUV infection in french forestry workers. Although the original sample population was selected to assess exposure to Borrelia burgdorferi they made an appropriate subject selection for this second study. The laboratory methods are well described and are correct.

R1 Point 1: I suggest giving more information in the introduction about prevalence studies in rodent reservoirs in the different regions of France; knowing the magnitude of the exposure help to understand the results better.

  • Authors: A systematic monitoring system for rodents (population density, hantavirus prevalence) does not exist in France. Studies on bank voles have been restricted to the endemic areas and as expected bank voles have been found infected in this areas, with level of IgG anti-PUUV prevalence varying largely according to time. We added this information to the introduction (lines 51-55).

R1 Point 2: It would also be informative to know if there are sequelae of HFRS concentrated in any particular group related to forestry workers. This information would give fundament to the main goal of the study. If there are no surveillance studies, propose them in the discussion.

  • Authors: Unfortunately we do not have data on sequelae of HFRS in any particular group among forestry workers. It was not planned to record this information in our study.

R1 Point 3: In addition, describe the dynamic of rodents in the areas where the researchers are interested and report if any human outbreak has occurred and where the associated risk factors are. Finally, the information obtained contributes to understanding this zoonosis in risk populations and promotes recognition and prevention.

  • Authors: The dynamic of rodents in the areas of interest for this work is presented in lines 46-51 in the introduction. The bank vole is a species typically living in areas of forest. It is widely distributed in metropolitan France, except on the Mediterranean coast. There is not a lot of data on the presence and density of bank voles in the territory nor on the prevalence of hantaviruses in this population, especially outside the areas known as endemic.
  • Authors: To the best of our knowledge, the risk factors are to be living in a house near the forest, presence of rodents in the dwellings, leisure or professional activities in the forest, as shown in this article https://www.bmj.com/content/318/7200/1737.short. We added lines 40-42 in the introduction to complete this information.

Reviewer 2 Report

The paper by Krug et al deals with recent hantavirus seroprevelance in areas of France. This paper establishes benchmarks for hantavirus seroprevelence, building on previous work by the authors. The subject is of value as recent work in this field has established that in contrast to many viral infections hantaviruses often remain geographically localised with their rodent reservoir, meaning that detailed mapping of seroprevelance is useful to establish risk stratification for occupational exposure. This paper contributes to the better understanding of this phenomenon. The study design is good and sufficient although as noted fewer samples were collected than planned. The authors could comment on why so few females were sampled, in regard to the known difference in pathology between hantavirus infection in men and women.

English useage is good with some minor problems (e.g. lines 217-221). 

Author Response

R2 : The paper by Krug et al deals with recent hantavirus seroprevelance in areas of France. This paper establishes benchmarks for hantavirus seroprevelence, building on previous work by the authors. The subject is of value as recent work in this field has established that in contrast to many viral infections hantaviruses often remain geographically localised with their rodent reservoir, meaning that detailed mapping of seroprevelance is useful to establish risk stratification for occupational exposure. This paper contributes to the better understanding of this phenomenon. The study design is good and sufficient although as noted fewer samples were collected than planned.

R2 Point 1: The authors could comment on why so few females were sampled, in regard to the known difference in pathology between hantavirus infection in men and women.

  • Authors: The reason why so few females were sampled was related to our sampling frame. Our target population were forestry workers, monitored by occupational health physicians of the agriculture social fund in 15 administrative regions in the northern half of France. Since the proportion of female forestry workers was low (7% [748/11,009]), that affected our final sample (which is representative of the target population). We added a sentence in the discussion on this limitation (lines 267-270).

R2 Point 2: English usage is good with some minor problems (e.g. lines 217-221). 

  • Authors: English usage was revised throughout the manuscript.